# COVID-19 severity and mortality in multiple sclerosis are not associated with immunotherapy: Insights from a nation-wide Austrian registry

**Gabriel Bsteh**[1]*, Hamid Assar[2], Harald Hegen[3], Bettina Heschl[4], Fritz Leutmezer[1], **Franziska Di Pauli**[3], Christiane Gradl[5], Gerhard Traxler[6], Gudrun Zulehner[1], **Paulus Rommer**[1], Peter Wipfler[7], Michael Guger[6], Christian Enzinger[4], Thomas Berger[1], **for the AUT-MuSC investigators**[1¶]

**1** Department of Neurology, Medical University of Vienna, Vienna, Austria, **2** Department of Neurology, Kepler University Hospital, Linz, Austria, **3** Department of Neurology, Medical University of Innsbruck, Innsbruck, Austria, **4** Department of Neurology, Medical University of Graz, Graz, Austria, **5** Department of Neurology, Medical University of St. Pölten, St. Pölten, Austria, **6** Department of Neurology 2, Med Campus III, Kepler University Hospital GmbH, Linz, Austria, **7** Department of Neurology, Paracelsus Medical University of Salzburg, Salzburg, Austria

¶ Membership of AUT-MuSC investigators is provided in the Acknowledgments.
* gabriel.bsteh@meduniwien.ac.at

**Data Availability Statement:** Data supporting the findings of this study are available from the corresponding author upon reasonable request by

## Abstract

### Background

The COVID-19 pandemic challenges neurologists in counselling patients with multiple sclerosis (pwMS) regarding their risk by SARS-CoV-2 and in guiding disease-modifying treatment (DMT).

### Objective

To characterize the prevalence and outcome of COVID-19 in pwMS specifically associated with different DMT in a nationwide population-based study.

### Methods

We included patients aged $\geq$18 years with a confirmed diagnosis of MS and a diagnosis of COVID-19 established between January 1, 2020 and December 31, 2020. We classified COVID-19 course as either mild, severe or fatal. Impact of DMT and specifically immunosuppressants (alemtuzumab, cladribine, fingolimod, ocrelizumab or rituximab) on COVID-19 outcome was determined by multivariable models, adjusted for a-priori-risk.

### Results

Of 126 MS patients with COVID-19 (mean age 43.2 years [SD 13.4], 71% female), 86.5% had a mild course, 9.5% a severe course and 3.2% died from COVID-19. A-priori-risk significantly predicted COVID-19 severity ($R^2$ 0.814; $p<0.001$) and mortality ($R^2$ 0.664; $p<0.001$). Adjusting for this a-priori-risk, neither exposure to any DMT nor exposure to specific

a qualified researcher and can be accessed from and upon approval by the ethics committee of the Medical University Vienna (contact via letter to Borschkegasse 9, 1090 Vienna, Austria) since data contain potentially sensitive information.

**Funding:** The author(s) received no specific funding for this work.

**Competing interests:** I have read the journal's policy and the authors of this manuscript have the following competing interests: Gabriel Bsteh: has participated in meetings sponsored by, received speaker honoraria or travel funding from Biogen, Celgene, Merck, Novartis, Roche, Sanofi-Genzyme and Teva, and received honoraria for consulting Biogen, Roche and Teva. Hamid Assar: has participated in meetings sponsored by, received honoraria (advisory boards, consultations) or travel funding from Biogen, Merck, Novartis, Roche, Sanofi-Genzyme, and Teva. Harald Hegen: has participated in meetings sponsored by, received speaker honoraria or travel funding from Bayer, Biogen, Merck, Novartis, Roche, Sanofi-Genzyme, Siemens and Teva, and received honoraria for consulting Biogen, Novartis and Teva. Bettina Heschl: has nothing to disclose. Fritz Leutmezer: has participated in meetings sponsored by or received honoraria for acting as an advisor/speaker for Bayer, Biogen, Celgene, MedDay, Merck, Novartis, Roche, Sanofi-Genzyme and Teva. Franziska Di Pauli: has participated in meetings sponsored by, received honoraria (lectures, advisory boards, consultations) or travel funding from Bayer, Biogen, Celgene, Merck, Novartis, Sanofi-Genzyme, Roche and Teva. Christiane Gradl: has participated in meetings sponsored by, received honoraria (lectures, consultations) and/or travel funding from Biogen, D-Pharma, Merck, Novartis, Roche, Sanofi-Genzyme, and Teva. Gerhard Traxler: has participated in meetings sponsored by, received honoraria (lectures, advisory boards, consultations) or travel funding from Biogen, Celgene, Merck, Novartis, Roche, Sanofi-Genzyme and Teva. Gudrun Zulehner: has participated in meetings sponsored by or received travel funding from Biogen, Merck, Novartis, Roche, Sanofi-Genzyme and Teva. Paulus Rommer: has received honoraria for consultancy/ speaking from AbbVie, Allmiral, Alexion, Biogen, Merck, Novartis, Roche, Sandoz, Sanofi Genzyme, has received research grants from Amicus, Biogen, Merck, Roche. Peter Wipfler: has received funding for travel and honoraria (lectures, advisory boards) from Bayer, Biogen, Celgene, Merck, Novartis, Roche, Sanofi-Genzyme and Teva. Michael Guger: has received support and honoraria for research, consultation, lectures and education from Almirall,

immunosuppressive DMT were significantly associated with COVID-19 severity (odds ratio [OR] 1.6; p = 0.667 and OR 1.9; p = 0.426) or mortality (OR 0.5; p = 0.711 and 2.1; 0.233) when compared to no DMT.

## Conclusions

In a population-based MS cohort, COVID-19 outcome was not associated with exposure to DMT and immunosuppressive DMT when accounting for other already known risk factors. This provides reassuring evidence that COVID-19 risk can be individually anticipated in MS and–except for a very small proportion of high-risk patients–treatment decisions should be primarily focused on treating MS rather than the pandemic.

## Introduction

The severe acute respiratory syndrome coronavirus 2 (SARS-CoV-2) pandemic has caused more than 83 million confirmed infections worldwide and approximately 1.8 million have died from the consequences of the virus-associated respiratory disease (CoronaVirus-Disease 2019, COVID-19) as by December 31st, 2020. Mortality and clinical severity of COVID-19 are strongly dependent of age and preexisting comorbidities [1, 2].

Multiple sclerosis (MS) is often considered as a disease affecting young adults, but a substantial number of patients with MS (pwMS) are older than 60 years and might thus be at an increased risk of severe morbidity and mortality from COVID-19 [2–4].

Further, there is particular concern whether immunomodulatory or immunosuppressive disease-modifying treatments (DMT), which are to some extent associated with a greater risk of infection, also increase the risk for COVID-19 severity and mortality [5, 6]. Although initial reports indicated a low proportion of pwMS at high risk of COVID-19 mortality, evidence regarding the effect of DMT on the course of COVID-19 is as scarce as direly needed for guiding pwMS through the pandemic [5, 7, 8].

The objective of this study was to characterize the prevalence, severity and overall mortality of SARS-CoV-2 infections in pwMS specifically associated with different DMT in a nationwide population-based study.

## Methods

### Patients and data collection

In this nationwide multicenter observational study, we included patients with a confirmed diagnosis of MS aged ≥18 years and with a diagnosis of COVID-19 (defined by either a positive SARS-CoV-2 polymerase chain reaction [PCR] or clinical diagnosis supported by i) a subsequent positive SARS-CoV-2 antibody test or b) a positive SARS-CoV-2 PCR in a close contact person of the patient) established between January 1, 2020 and December 31, 2020 into the Austrian MS-COVID-19 registry (AUT-MuSC) [9].

Patients were recruited through the Austrian MS network, which is a collaboration of MS centers certified by the Austrian Neurological Society adhering to a common and controlled high-quality standard of managing and documenting about 13.500 patients with MS in Austria [10]. The study was designed and conducted in accordance with the Declaration of Helsinki, the General Data Protection Regulation and the Strengthening Reporting of Observational

Bayer, Biogen, Celgene, Genzyme, MedDay, Merck, Novartis, Octapharma, Roche, Sanofi-Genzyme, Shire and Teva. Christian Enzinger: has received funding for travel and speaker honoraria from Bayer, Biogen, Merck, Novartis, Roche, Sanofi-Genzyme, Shire and Teva. has received research support from Biogen, Celgene, Merck, and Teva; is serving on scientific advisory boards for Bayer, Biogen, Celgene, Merck, Novartis, Roche and Teva. Thomas Berger: has participated in meetings sponsored by and received honoraria (lectures, advisory boards, consultations) from pharmaceutical companies marketing treatments for MS: Allergan, Bayer, Biogen, Bionorica, Celgene, MedDay, Merck, Novartis, Octapharma, Roche, Sanofi-Genzyme, Teva. His institution has received financial support in the past 12 months by unrestricted research grants (Bayer, Biogen, Merck, Novartis, Sanofi Aventis, Teva) and for participation in clinical trials in multiple sclerosis sponsored by Alexion, Bayer, Biogen, Merck, Novartis, Octapharma, Roche, Sanofi-Genzyme, Teva. This does not alter our adherence to PLOS ONE policies on sharing data and materials.

Studies in Epidemiology (STROBE) guidelines and was approved by the ethics committee of the Medical University Vienna (ethical approval number: EK 1338–2020).

When the treating neurologist was informed about a diagnosis of COVID-19 or a confirmed SARS-CoV-2 infection in an MS patient either at an on-site visit or remotely, data were collected retrospectively by fulfilling a pseudonymized case report form from a review of medical records, which was then sent to the coordinating study center at the Department of Neurology, Medical University of Vienna. Patients included were informed about the objective of the study and written informed consent was obtained.

Data collected included demographic data, details of MS course, DMT history, a detailed documentation of prior and current comorbidities and a detailed description of source, course, available laboratory and radiographic diagnostics, treatment and outcome of COVID-19.

## Definitions and endpoints

Patients were classified regarding their a-priori risk of COVID-19 severity and mortality according to a recently developed risk score (MS-COV-risk, see Table 1), categorizing MS patients based on age, physical disability (Expanded Disability Status Scale [EDSS] score), smoking status, obesity (body-mass-index $\geq$30), and presence of cardiovascular disease (coronary heart disease and/or ischemic heart failure and/or cardiac valve disease), chronic pulmonary disease (asthma, obstructive pulmonary disease [COPD] or pulmonary fibrosis), diabetes mellitus, chronic kidney disease and current malignancy as having either low (<1%), mild (<5%), moderate (~15%), high (~30%) or very high risk (~50%) of COVID-19 mortality [7].

Patients were grouped according to their DMT status at the time of SARS-CoV-2 infection as either receiving no DMT (N-DMT); immunomodulating DMT (IM-DMT) comprising dimethyl fumarate, glatiramer acetate, interferon-beta preparations, natalizumab, and teriflunomide; or immunosuppressive DMT (IS-DMT) comprising alemtuzumab, cladribine, fingolimod, ocrelizumab or rituximab [8, 11].

**Table 1. MS-COV-risk score.**

| Factor | Score |
|---|---|
| Diabetes AND age <40 years | 5 |
| Age $\geq$65 years | 3 |
| Chronic kidney disease | 3 |
| Chronic obstructive pulmonary disease | 1 |
| Cardiovascular disease | 1 |
| Current Malignancy | 1 |
| Obesity (BMI $\geq$30) | 1 |
| Diabetes | 1 |
| Smoking | 1 |
| Severe physical disability (EDSS >6) | 1 |
| Age <40 | -6 |
| **Risk category** | **Score Interval** |
| Low risk | $\leq$0 |
| Mild risk | 1–3 |
| Moderate risk | 4–7 |
| High risk | 8–11 |
| Very high risk | $\geq$12 |

BMI: body mass index. EDSS: expanded disability status scale. See reference [7].

The primary endpoint was COVID-19 severity defined as the clinical status at the most severe point of COVID-19 course on a 4-point ordinal scale, where 0 indicates an asymptomatic course; 1 a mild course (no pneumonia or mild pneumonia without hospitalization); 2 a severe course requiring hospitalization and fulfilling at least one of five criteria (breathing rate >30/minute, SpO2 ≤93%, PaO2/FiO2-Ratio <300, Pulmonary infiltrate >50% within 24-48h, requirement of noninvasive ventilation, high-flow oxygen, mechanical ventilation or extracorporeal membrane oxygenation), and 3 death. The secondary endpoint was defined as mortality from COVID-19.

## Statistical analysis

Statistical analysis was performed using SPSS 26.0 (SPSS Inc, Chicago, IL). Categorical variables were expressed in frequencies and percentages. Continuous variables were tested for normal distribution by Shapiro-Wilk test and expressed as mean and standard deviation (SD) or median and range as appropriate. Univariate group comparisons were conducted by t-test, ANOVA, Mann-Whitney-U test, Kruskal-Wallis test or chi-square test as appropriate.

First, we investigated the strength of association between a-priori risk scoring and COVID-19 outcome (severity and mortality) by performing univariable correlation analyses (Spearman-rho) and multivariable logistic regression models with COVID-19 severity/mortality as the dependent variable and MS-COV-risk score (absolute score and risk categories) as the independent variable adjusting for sex (age is already included in the MS-COV-risk score) and lymphopenia before COVID-19 as a possible risk factor not included in the MS-COV-risk score [8]. Then, we tested the independent impact of overall DMT as well as IM-DMT and IS-DMT on COVID-19 outcome by calculating multivariate logistic regression models with COVID-19 severity/mortality as the dependent variable and DMT categories (reference category: N-DMT) as the independent variable adjusted for a-priori-risk as expressed by MS-COV-risk score adjusting for sex and lymphopenia before COVID-19. Finally, we conducted sensitivity analyses evaluating the robustness of results to the impact of any single DMT substance by stepwise removal from analyses. Robustness of the statistically significant differences to unidentified confounders not accounted for by MS-COV-risk score was quantified with Rosenbaum sensitivity test for Hodges–Lehmann Γ [12]. Missing values were handled by multiple (20 times) imputation using the missing not at random (MNAR) approach with pooling of estimates according to Rubin's rules [13]. A two-sided p-value <0.05 was considered statistically significant.

## Results

We included 126 patients with a mean age of 43.2 years [SD 13.4] and a female predominance of 71%. Overall characteristics of the study cohort are given in Table 2. A detailed description of every patient included is presented in (S1 Table).

The number of comorbidities associated with increased COVID-19 morbidity (cardiovascular disease, chronic obstructive pulmonary disease, chronic kidney disease, diabetes and concurrent malignancy) as well as MS-associated physical disability (EDSS) significantly increased with age (rho = 0.176, p = 0.049 and rho = 0.576, p<0.001, respectively). Frequency of obesity and smoking was not associated with age, but pwMS receiving DMT were significantly younger than those without DMT (mean age 39.0 vs. 53.6; p<0.001). Of the 103 patients with available differential blood count before COVID-19 onset, 19 (18.4%) had lymphopenia, with 7 (6.8%) grade 3 or lower.

Overall, 5 (4.0%) were asymptomatically infected with SARS-CoV-2, 109 (86.5%) had mild COVID-19 and 12 (9.5%) had a severe course, of whom 4 (3.2%) died from COVID-19.

**Table 2. Characteristics of the AUT-MuSC-19 cohort.**

|  | n = 126 |
|---|---|
| Female[a] | 90 (71.4) |
| Age (years)[b,c] | 43.2 (13.4; 21–79) |
| BMI [c] | 24.1 (17.4–41.0) |
| Smokers[a] | 17 (3.5) |
| Ethnicity[a] |  |
| Caucasian[a] | 123 (97.6) |
| Other[a] | 3 (2.4) |
| No of comorbidities associated with increased COVID-19 morbidity[c*] | 0 (0–5) |
| Disease duration (years) [b] | 12.0 (9.3) |
| Disease course[a] |  |
| RRMS[a] | 98 (77.8) |
| SPMS[a] | 19 (15.1) |
| PPMS[a] | 9 (7.1) |
| EDSS[c] | 2.0 (0–8.5) |
| On DMT[a] | 90 (71.4) |
| IM-DMT | 48 (38.1) |
| Interferon-beta[a] | 6 (4.8) |
| Glatiramer acetate[a] | 11 (8.7) |
| Dimethyl fumarate[a] | 19 (15.1) |
| Teriflunomide[a] | 2 (1.6) |
| Natalizumab[a] | 10 (7.9) |
| IS-DMT | 41 (32.5) |
| Fingolimod[a] | 16 (12.7) |
| Ocrelizumab/Rituximab[a] | 12 (9.5) |
| Alemtuzumab[a] | 2 (1.6) |
| Cladribine | 2 (1.6) |
| Azathioprin[a] | 1 (0.8) |
| Lymphopenia at last lab before SARS-CoV2 infection[a**] | 19 (18.4) |
| Grade 3 or lower[a] | 7 (6.8) |

BMI: body mass index. DMT: disease modifying treatment. EDSS: Expanded Disability Status Scale. IM-DMT: Immunomodulating DMT = dimethyl fumarate, glatiramer acetate, interferon beta preparations, natalizumab, and teriflunomide. IS-DMT: Immunosuppressive DMT = alemtuzumab, cladribine, fingolimod, ocrelizumab or rituximab. MS: multiple sclerosis. PPMS: primary progressive MS. RRMS: relapsing-remitting MS. SPMS: secondary progressive MS.

[a] absolute number and percentage.

[b] mean and standard deviation.

[c] median and minimum-maximum range.

*defined as cardiovascular diseases, chronic obstructive pulmonary disease, chronic kidney disease, diabetes and concurrent malignancy.

**available from 103 patients.

## A-priori-risk and COVID-19 outcome

Overall, a-priori risk categories were distributed as follows: 75 (59.5%) low risk, 39 (3.0%) mild risk, 8 (6.3%) moderate risk and 4 (3.2%) high risk of COVID-19 mortality. We therefore condensed patients with moderate and high risk into one group to facilitate further categorical analyses.

MS-COV-risk score was strongly correlated with COVID-19 outcome (rho = -0.426, p<0.001). Among the 103 patients for whom lymphocyte counts were available before COVID-19 onset, we did not find an association between lymphopenia and COVID-19 severity or mortality.

In the multivariable models adjusting for sex and lymphopenia, MS-COV-risk score significantly predicted COVID-19 severity (odds ratio [OR]: 1.4 per 1 point increase; 95% confidence interval [CI]: 1.2–3.7; p = 0.001; $R^2$ 0.814; p<0.001) and mortality (OR: 2.4; CI: 1.2–5.4; p = 0.007; $R^2$ 0.664; p<0.001).

### The impact of DMT on COVID-19 outcome

COVID-19 outcome according to DMT categories and single DMT substances is shown in Fig 1. We did not find any significant association between DMT and COVID-19 morbidity, neither overall, nor when stratifying according to a-priori risk categories (Table 3).

Adjusting for MS-COV-risk score, sex and lymphopenia before SARS-CoV-2 infection, exposure to any DMT was neither associated with COVID-19 severity (OR: 1.6; CI: 0.2–11.9; p = 0.667) nor with mortality (OR: 0.5; CI: 0.2–19.6; p = 0.711, Table 4) when compared to no DMT. Similarly, exposure to IS-DMT was not associated with COVID-19 severity (OR: 1.9; CI: 0.5–9.4; p = 0.426) and mortality (OR 2.1; CI: 0.8–12.3; p = 0.233).

## Discussion

In this nationwide population-based study of SARS-CoV-2 infections in pwMS, we report several key findings: 1) prevalence, severity and mortality in pwMS in Austria are comparable to the general population, 2) COVID-19 severity and mortality can be predicted by applying a score of known a-priori risk factors and 3) COVID-19 severity and mortality are not associated with exposure to DMT and specifically immunosuppressive DMT when accounting for a-priori risk.

### Prevalence

Given a population of approximately 13,500 MS patients in Austria, the 126 pwMS included in this population-based study would translate to a prevalence of 0.93% at the end of 2020.[10] This would seem well below the prevalence of 4.2% of COVID-19 in the general population (360,815 confirmed cases in 8.5 million people) during the same period [14, 15]. However, the prevalence in the general population is based on the number of positive SARS-CoV-2 tests

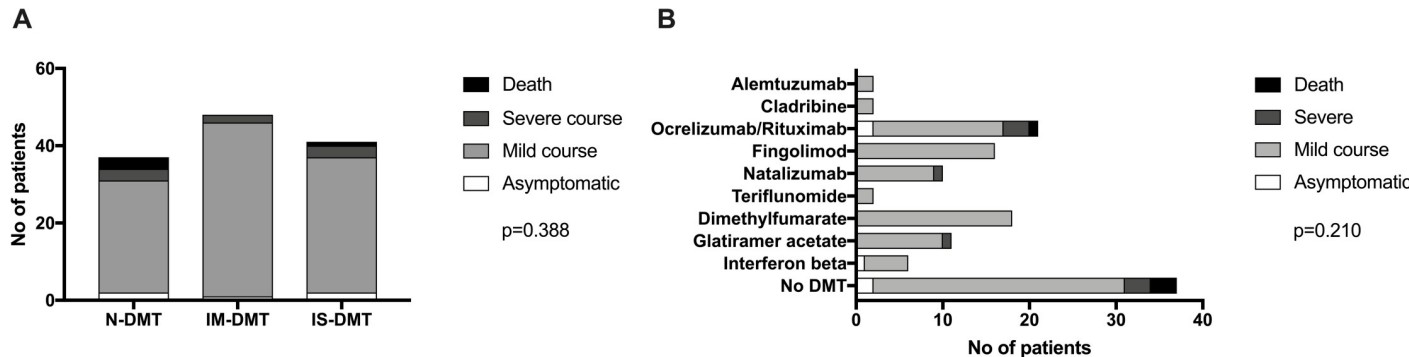

**Fig 1.** COVID-19 outcome according to DMT classes (A) and single substances (B). DMT: disease modifying treatment. IM-DMT: Immunomodulating DMT = dimethyl fumarate, glatiramer acetate, interferon beta preparations, natalizumab, and teriflunomide. IS-DMT: Immunosuppressive DMT = alemtuzumab, cladribine, fingolimod, ocrelizumab or rituximab. p-values calculated by chi-square test. N-DMT: no DMT.

**Table 3. COVID-19 severity and mortality according to a-priori risk and DMT class.**

| Risk category | | At risk | Severe COVID-19 | p-value | Fatal COVID-19 | p-value |
|---|---|---|---|---|---|---|
| Low risk (Score ≤0) N = 75 | N-DMT | 17 | 0 | 0.455[1] | 0 (0) | NA |
| | IM-DMT | 31 | 1 | | 0 (0) | |
| | IS-DMT | 27 | 2 | | 0 (0) | |
| Mild risk (Score 1–3) N = 39 | N-DMT | 11 | 0 | 0.315[1] | 0 (0) | NA |
| | IM-DMT | 16 | 0 | | 0 (0) | |
| | IS-DMT | 12 | 1 | | 0 (0) | |
| Moderate/high risk (Score ≥4) N = 12 | N-DMT | 9 | 6 | 0.687[1] | 3 | 0.677[1] |
| | IM-DMT | 1 | 1 | | 0 | |
| | IS-DMT | 2 | 1 | | 1 | |

DMT: disease modifying treatment. IM-DMT: Immunomodulating DMT = dimethyl fumarate, glatiramer acetate, interferon beta preparations, natalizumab, and teriflunomide. IS-DMT: Immunosuppressive DMT = alemtuzumab, cladribine, fingolimod, ocrelizumab or rituximab.

[1]calculated by chi-square test. N-DMT: no DMT.

rather than symptomatic COVID-19. Considering that only about 20% of seropositive MS patients appear to have symptomatic COVID-19 and that only 5/126 (3.9%) patients were asymptomatic in our cohort, the true prevalence of SARS-CoV-2 infection in the Austrian MS population can be estimated at 4.1% and, thus, lies well within the range of the general population [16]. This is in line with previous studies applying various definitions of SARS-CoV-2 infection/COVID-19 and pharmacoepidemiological techniques adding to the mounting evidence that pwMS are neither more nor less likely to contract SARS-CoV-2 [17–19]. Thus, it can be inferred that the AUT-MuSC-19 registry is likely to include most of pwMS with a symptomatic SARS-CoV-2 infections in Austria and is likely representative of a central European, primarily Caucasian MS population.

## COVID severity and mortality

Evidence regarding the specific morbidity and mortality from COVID-19 in pwMS is still scarce. In our population-based cohort, we found 9.5% with severe COVID-19 course and 3.2% mortality. Encouragingly, this is within the range of mortality reports both in Austria (1.7%; 6,222 of 360,815 confirmed cases) and globally (2.2%) in the general population and in line with previous studies suggesting that MS patients do not have an increased risk of COVID-19 fatality compared with the population at large [8, 14, 20–22].

**Table 4. COVID-19 severity and mortality depend on a priori risk but not DMT class.**

| COVID-19 severity | | | | COVID-19 mortality | | | |
|---|---|---|---|---|---|---|---|
| | OR | 95% CI | p value | | OR | 95% CI | p value |
| **MS-COV-risk score (per point)** | **1.5** | **1.2–3.7** | **0.005** | **MS-COV-risk score (per point)** | **2.4** | **1.2–7.3** | **0.021** |
| DMT[1] | 1.6 | 0.2–11.9 | 0.667 | DMT[1] | 0.5 | 0.2–19.6 | 0.711 |
| IM-DMT | 1.1 | 0.2–7.3 | 0.943 | IM-DMT | 0.2 | 0.0–3.4 | 0.122 |
| IS-DMT | 1.9 | 0.5–9.4 | 0.426 | IS-DMT | 2.1 | 0.8–12.3 | 0.233 |
| | R square 0.832; p<0.001 | | | | R square 0.683; p<0.001 | | |

[1]reference category: no DMT.

IM-DMT: Immunomodulating DMT = dimethyl fumarate, glatiramer acetate, interferon beta preparations, natalizumab, and teriflunomide. IS-DMT: Immunosuppressive DMT = alemtuzumab, cladribine, fingolimod, ocrelizumab or rituximab. OR: odds ratio. 95% CI: confidence interval. Calculated by binary linear regression models with severe COVID-19 and fatal COVID-19 adjusted for sex and lymphopenia before COVID-19.

It is well established that in the general population COVID-19 severity and mortality increases with older age and in the presence of comorbidities, i.e. cardiovascular disease, pulmonary disease, chronic kidney disease, malignancy, obesity and smoking [1, 2, 23, 24]. In MS patients, advanced physical disability represents an additional factor associated with poor COVID-19 outcome [8, 22]. We have recently introduced a score (MS-COV-risk) to cumulatively quantify these a-priori risk factors in order to predict COVID-19 severity and mortality in pwMS [7]. Here, we found that by applying the MS-COV-risk score, 81% of the variation in COVID-19 severity ($R^2$ 0.862; p<0.001) and 66% of mortality ($R^2$ 0.663; p<0.001) could be predicted. This both validates the predictive ability of the MS-COV-risk score and confirms that COVID morbidity in pwMS is largely determined by factors independent of MS and, thus, can be anticipated.

## The role of DMT

One of the most pressing questions in managing pwMS during the pandemic is whether immunomodulatory or immunosuppressive DMT increase the risk for COVID-19 severity and mortality. While MS is generally not associated with increased morbidity from viral pathogens, some DMTs are to some extent associated with a greater risk of infection [21]. Therefore, initial recommendations of various expert committees at the beginning of the pandemic in early 2020 have offered very conservative advice with some even suggesting discontinuation of DMT. As the number of cases grew over the subsequent months, evidence is reassuringly growing that poor outcome in pwMS contracting COVID-19 is primarily dependent on age, obesity, comorbidities and degree of physical disability rather than DMT use in general [5, 8, 22, 25–28]. Our data confirm and extend these findings showing that when accounting for a-priori risk, COVID-19 severity and mortality are independent of both overall exposure to DMT as well immunosuppressive DMT. This has also been recently shown in a large pharmacoepidemiological study [18]. However, there is some concern regarding a possibly increased risk under treatment with anti-CD20 B cell depleting agents as indicated by a recent large observational study [28]. Also, there have been reports of a decreased serologic response in pwMS after SARS-CoV-2 infection [29, 30]. Our study is not sufficiently powered to determine the effect of single DMT substances on COVID outcome. However, sensitivity analyses did not indicate a significant change of results when removing single DMT substances. Similar to other studies, we also did not find an association between lymphopenia and COVID-19 morbidity, neither in univariable analyses nor when accounting for a-priori risk and DMT [8, 26].

Therefore, in pwMS with low to moderate risk of COVID-19 mortality, the benefit-risk ratio is clearly in favor of both continuing and initiating DMT when indicated by the MS course in the respective individual patient. The majority of the small group of pwMS displaying a high a-priori risk of having severe COVID-19 does not even receive DMT (in our cohort 0 of 4 patients) and is unlikely to display significant disease activity, hence, the question of stopping or delaying DMT hardly arises [7]. Consequently, most expert committees have now adopted less cautious guidelines emphasizing the paramount need for ensuring optimal treatment of MS individually integrating MS specific parameters as well as comorbidities, social circumstances, personal risk perception etc., especially during the pandemic [31, 32]. In this light, reports of significant drops in patient rate and changed/deferred DMT regimens are particularly concerning as quality of life of pwMS greatly depends on prompt access to a broad range of health and care service [33, 34]. It is essential for MS caregivers to uphold the standard of care for pwMS, including continuation of safety monitoring procedures as far as possible depending on local circumstances [21, 32].

### Strengths and limitations

The main strengths of this study are its population-based approach and the detailed characterization of the study cohort provided by the high-quality data from certified specialized MS centers. We have to acknowledge some potential limitations inherent to the study design.

MS patients with asymptomatic SARS-CoV-2 infection may have been systematically missed due to the study design. As already noted earlier, our study is not sufficiently powered to determine the effect of single DMT substances on COVID outcome. Thus, we had to use a categorization in immunomodulating and immunosuppressive DMTs in the primary analyses, resulting in an estimated power of 73% for detecting an increased risk of COVID severity by an odds ratio of 2. However, we conducted sensitivity analyses evaluating the robustness of results to the impact of any single DMT substance by stepwise removal, which did not indicate a significant change of results. In addition, there may be a referral bias as severe COVID-19 courses may be more likely to be reported. On the other hand, patients with advanced and progressive MS or patients not receiving DMT are less frequently seeing a neurologist regularly and, thus, this cohort might be underrepresented in this study. There may also be confounders influencing COVID severity/mortality in pwMS unaccounted for by MS-COV-risk score and DMT. However, Rosenbaum bounds did indicate only a small potential impact of hidden bias not accounted for in the multivariable models [12].

## Conclusion

We showed in a population-based MS cohort that COVID-19 severity and mortality are not associated with exposure to DMT and immunosuppressive DMT when accounting for unmodifiable risk factors. This provides reassuring evidence that the COVID-19 risk can be anticipated in MS and–except for a very small proportion of high-risk patients–treatment decisions should be primarily focused on treating MS rather than the pandemic.

## Supporting information

**S1 Table. Characteristics of all 126 patients included.** ATZ: alemtuzumab, CHD: coronary heart disease., CKD: Chronic kidney disease, CLA: cladribine, COPD: chronic obstructive pulmonary disease, DMF: dimethyl fumarate, DMT: disease modifying treatment, EDSS: Expanded disability status scale, FTY: fingolimod, F: female, GLA: glatiramer acetate, IFN: interferon beta, M: male, NTZ: natalizumab, OCR: ocrelizumab, PCR: SARS-CoV-2-polymerase-chain-reaction, RTX: rituximab, TERI: teriflunomide.
(DOCX)

## Acknowledgments

We thank all the AUT-MuSC-19 investigators, clinical research staff, and especially the patients for helping to collect these data. The named individuals were not compensated for their help. Lead AUT-MuSC investigator: Gabriel Bsteh, MD, PhD; Email: gabriel.bsteh@meduniwien.ac.at.

AUT-MuSC investigators in alphabetical order: Assar, Hamid (Kepler University Hospital, Linz, Austria); Berger, Thomas (Medical University of Vienna, Vienna, Austria); Böck, Klaus (Kepler University Hospital, Linz, Austria); Bsteh, Christian (Private practice neurologist, Salzburg, Austria); Bsteh, Gabriel (Medical University of Vienna, Vienna, Austria); Di Pauli, Franziska (Medical University of Innsbruck, Innsbruck, Austria); Enzinger, Christian (Medical University of Graz, Graz, Austria); Gradl, Christiane (Medical University of St. Pölten, St. Pölten, Austria); Guger, Michael (Med Campus III, Kepler University Hospital GmbH,

Linz, Austria); Hegen, Harald (Medical University of Innsbruck, Innsbruck, Austria); Heschl, Bettina (Medical University of Graz, Graz, Austria); Hiller, Marie-Sophie (Barmherzige Brüder Hospital, Eisenstadt, Austria); Kornek, Barbara (Medical University of Vienna, Vienna, Austria); Leutmezer, Fritz (Medical University of Vienna, Vienna, Austria); Mayr, Markus (District Hospital Kufstein, Kufstein, Austria); Morgenstern, Gabriele (Private practice neurologist, Lienz, Austria); Rommer, Paulus (Medical University of Vienna, Vienna, Austria); Schnabl, Peter (Private practice neurologist, Velden, Austria); Schneider-Koch, Gabriela (Ottakring Hospital, Vienna, Austria); Schrotter, Gabriele (LKH West Graz, Graz, Austria); Traxler, Gerhard (Clinic for Neurology 2, Med Campus III, Kepler University Hospital GmbH, Linz, Austria); Wipfler, Peter (Paracelsus Medical University of Salzburg, Salzburg, Austria); Zulehner, Gudrun (Medical University of Vienna, Vienna, Austria); Zrzavy, Tobias (Medical University of Vienna, Vienna, Austria).

## Author Contributions

**Conceptualization:** Gabriel Bsteh, Thomas Berger.

**Data curation:** Gabriel Bsteh, Hamid Assar, Harald Hegen, Bettina Heschl, Fritz Leutmezer, Franziska Di Pauli, Christiane Gradl, Gerhard Traxler, Gudrun Zulehner, Paulus Rommer, Peter Wipfler, Michael Guger, Christian Enzinger.

**Formal analysis:** Gabriel Bsteh.

**Methodology:** Hamid Assar, Harald Hegen, Bettina Heschl, Fritz Leutmezer, Franziska Di Pauli, Christiane Gradl, Gerhard Traxler, Gudrun Zulehner, Paulus Rommer, Peter Wipfler, Michael Guger, Christian Enzinger.

**Supervision:** Thomas Berger.

**Writing – original draft:** Gabriel Bsteh.

**Writing – review & editing:** Hamid Assar, Harald Hegen, Bettina Heschl, Fritz Leutmezer, Franziska Di Pauli, Christiane Gradl, Gerhard Traxler, Gudrun Zulehner, Paulus Rommer, Peter Wipfler, Michael Guger, Christian Enzinger, Thomas Berger.

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
