## [Decision Letter · Decision Letter 0]

25 May 2021

PONE-D-21-12442

COVID-19 severity and mortality in multiple sclerosis do not depend on immunotherapy: insights from a nation-wide Austrian registry

PLOS ONE

Dear Dr. Bsteh,

Thank you for submitting your manuscript to PLOS ONE. After careful consideration, we feel that it has merit but does not fully meet PLOS ONE’s publication criteria as it currently stands. Therefore, we invite you to submit a revised version of the manuscript that addresses the points raised during the review process.

Please refer to minor issues raised by referee #1. For referee #2, please address formal issues as outlined below..

We look forward to receiving your revised manuscript.

Kind regards,

Orhan Aktas, M.D.

Academic Editor

PLOS ONE

Journal Requirements:

4. One of the noted authors is a group or consortium [AUT-MuSC investigators]. In addition to naming the author group, please list the individual authors and affiliations within this group in the acknowledgments section of your manuscript. Please also indicate clearly a lead author for this group along with a contact email address.

"I have read the journal's policy and the authors of this manuscript have the following competing interests:

Gabriel Bsteh: has participated in meetings sponsored by, received speaker honoraria or travel funding from Biogen, Celgene, Merck, Novartis, Roche, Sanofi-Genzyme and Teva, and received honoraria for consulting Biogen, Roche and Teva.

Hamid Assar: has participated in meetings sponsored by, received honoraria (advisory boards, consultations) or travel funding from Biogen, Merck, Novartis, Roche, Sanofi-Genzyme, and Teva.

Harald Hegen: has participated in meetings sponsored by, received speaker honoraria or travel funding from Bayer, Biogen, Merck, Novartis, Roche, Sanofi-Genzyme, Siemens and Teva, and received honoraria for consulting Biogen, Novartis and Teva.

Bettina Heschl: has nothing to disclose.

Fritz Leutmezer: has participated in meetings sponsored by or received honoraria for acting as an advisor/speaker for Bayer, Biogen, Celgene, MedDay, Merck, Novartis, Roche, Sanofi-Genzyme and Teva.

Franziska Di Pauli: has participated in meetings sponsored by, received honoraria (lectures, advisory boards, consultations) or travel funding from Bayer, Biogen, Celgene, Merck, Novartis, Sanofi-Genzyme, Roche and Teva.

Christiane Gradl: has participated in meetings sponsored by, received honoraria (lectures, consultations) and/or travel funding from Biogen, D-Pharma, Merck, Novartis, Roche, Sanofi-Genzyme, and Teva.

Gerhard Traxler: has participated in meetings sponsored by, received honoraria (lectures, advisory boards, consultations) or travel funding from Biogen, Celgene, Merck, Novartis, Roche, Sanofi-Genzyme and Teva.

Gudrun Zulehner: has participated in meetings sponsored by or received travel funding from Biogen, Merck, Novartis, Roche, Sanofi-Genzyme and Teva.

Paulus Rommer: has received honoraria for consultancy/speaking from AbbVie, Allmiral, Alexion, Biogen, Merck, Novartis, Roche, Sandoz, Sanofi Genzyme, has received research grants from Amicus, Biogen, Merck, Roche.

Peter Wipfler: has received funding for travel and honoraria (lectures, advisory boards) from Bayer, Biogen, Celgene, Merck, Novartis, Roche, Sanofi-Genzyme and Teva.

Michael Guger: has received support and honoraria for research, consultation, lectures and education from Almirall, Bayer, Biogen, Celgene, Genzyme, MedDay, Merck, Novartis, Octapharma, Roche, Sanofi-Genzyme, Shire and Teva.

Christian Enzinger: has received funding for travel and speaker honoraria from Bayer, Biogen, Merck, Novartis, Roche, Sanofi-Genzyme, Shire and Teva. has received research support from Biogen, Celgene, Merck, and Teva; is serving on scientific advisory boards for Bayer, Biogen, Celgene, Merck, Novartis, Roche and Teva.

Thomas Berger: has participated in meetings sponsored by and received honoraria (lectures, advisory boards, consultations) from pharmaceutical companies marketing treatments for MS: Allergan, Bayer, Biogen, Bionorica, Celgene, MedDay, Merck, Novartis, Octapharma, Roche, Sanofi-Genzyme, Teva. His institution has received financial support in the past 12 months by unrestricted research grants (Bayer, Biogen, Merck, Novartis, Sanofi Aventis, Teva) and for participation in clinical trials in multiple sclerosis sponsored by Alexion, Bayer, Biogen, Merck, Novartis, Octapharma, Roche, Sanofi-Genzyme, Teva."

Reviewers' comments:

Reviewer's Responses to Questions

**Comments to the Author**

1. Is the manuscript technically sound, and do the data support the conclusions?

Reviewer #1: Partly

Reviewer #2: Yes

2. Has the statistical analysis been performed appropriately and rigorously? 

Reviewer #1: Yes

Reviewer #2: Yes

3. Have the authors made all data underlying the findings in their manuscript fully available?

Reviewer #1: Yes

Reviewer #2: Yes

4. Is the manuscript presented in an intelligible fashion and written in standard English?

Reviewer #1: Yes

Reviewer #2: Yes

5. Review Comments to the Author

Reviewer #1: Bsteh et al present a timely, population-based study regarding COVID-19 severity and mortality in multiple sclerosis patients in Austria. The authors included 129 multiple sclerosis patients with COVID-19, 86.5% had a mild course, 9.5% a severe course and 3.2% died. According to the authors, COVID-19 prevalence of the study cohort lies well within the general population. Neither exposure to any diseases-modifying treatment nor exposure to specific immunosuppressive DMT were significantly associated with COVID-19 severity. The mansucript is well-written and the results are of interest. However, the manuscript would improve when addressing a few minor issues.

1. Methods/results and abstract/conclusions are slightly imbalanced. Correlation does not imply causation, and both the manuscript title and the conclusion "treatment decisions should be focused on treating MS rather than the pandemic" are rather strong for a population-based study (with power issues) including 'only' 129 multiple sclerosis patients. Furthermore, the authors should clearly state that (asymptomatic) COVID-19 patients with multiple sclerosis may have been systematically missed due to the study design. Please adapt accordingly.

2. In the methods section, the authors wrote that the reference category is "multiple sclerosis patients without disease-modifying treatment exposure". In order to increase readability, please mention the reference category also in the abstract, results and discussion when appropriate.

3. The authors state that any disease-modifying treatment exposure was not associated with COVID-19 severity. However, a recently published study suggests that interferon antibodies are present in COVID-19 patients with a life-threatening course, especially in men and older patients (DOI: 10.1126/science.abd4585). Therefore, substituting interferon(-beta) may be beneficial regarding COVID-19 severitiy. If feasible, the authors could carefully comment on the 6 (4.8%) multiple sclerosis patients receiving interferon-beta having mild (and not life-threatening) COVID-19 (according to Suppl Table 1).

I thank the authors for their relevant scientific work.

Reviewer #2: The authors report the findings on COVID-19 severity and mortality in MS patients investigated in a nation-wide Austrian registry. According to their results, outcome of COVID-19 is not modified by disease-modifying immunotherapy for MS.

The methods are sound, the results are clearly presented, and the conclusions are justified. I have only a few suggestions:

1) Methods. Please explain how diagnosis of COVID-19 was performed for the Austrian MS-COVID-19 registry.

2) Methods. Please insert a table outlining the MS-COV-risk score recently developed by the authors. This would help a lot to better understand the findings.

3) Table 1. please also show range for age

4) Page 6, punctuation after reference [12]

6. PLOS authors have the option to publish the peer review history of their article (what does this mean?). If published, this will include your full peer review and any attached files.

Reviewer #1: No

Reviewer #2: No

---

## [Author Response · Author response to Decision Letter 0]

23 Jun 2021

Rebuttal to comments of reviewers and editor from initial submission (PONE-D-21-12442: COVID-19 severity and mortality in multiple sclerosis do not depend on immunotherapy: insights from a nation-wide Austrian registry)

Journal Requirements

1. When submitting your revision, we need you to address these additional requirements.Please ensure that your manuscript meets PLOS ONE's style requirements, including those for file naming. 

Response: Done.

Response: Done.

Response: Done.

4. One of the noted authors is a group or consortium [AUT-MuSC investigators]. In addition to naming the author group, please list the individual authors and affiliations within this group in the acknowledgments section of your manuscript. Please also indicate clearly a lead author for this group along with a contact email address.

Response: Done.

5. Thank you for stating the following in the Competing Interests section: "I have read the journal's policy and the authors of this manuscript have the following competing interests. Please confirm that this does not alter your adherence to all PLOS ONE policies on sharing data and materials, by including the following statement: "This does not alter our adherence to PLOS ONE policies on sharing data and materials.” (as detailed online in our guide for authors http://journals.plos.org/plosone/s/competing-interests). If there are restrictions on sharing of data and/or materials, please state these. Please note that we cannot proceed with consideration of your article until this information has been declared. Please include your updated Competing Interests statement in your cover letter; we will change the online submission form on your behalf.

Response: Done.

Reviewers' comments

Reviewer: 1

Comment: Bsteh et al present a timely, population-based study regarding COVID-19 severity and mortality in multiple sclerosis patients in Austria. The authors included 129 multiple sclerosis patients with COVID-19, 86.5% had a mild course, 9.5% a severe course and 3.2% died. According to the authors, COVID-19 prevalence of the study cohort lies well within the general population. Neither exposure to any diseases-modifying treatment nor exposure to specific immunosuppressive DMT were significantly associated with COVID-19 severity. The mansucript is well-written and the results are of interest. However, the manuscript would improve when addressing a few minor issues.

Response: Thank you for this positive assessment of our study.

Comment: 1. Methods/results and abstract/conclusions are slightly imbalanced. Correlation does not imply causation, and both the manuscript title and the conclusion "treatment decisions should be focused on treating MS rather than the pandemic" are rather strong for a population-based study (with power issues) including 'only' 129 multiple sclerosis patients. Furthermore, the authors should clearly state that (asymptomatic) COVID-19 patients with multiple sclerosis may have been systematically missed due to the study design. Please adapt accordingly.

Response: Thank you for this comment. We agree and have made an effort to improve the balance of the manuscript (see title, abstract and discussion section)

Comment: 2. In the methods section, the authors wrote that the reference category is "multiple sclerosis patients without disease-modifying treatment exposure". In order to increase readability, please mention the reference category also in the abstract, results and discussion when appropriate.

Response: Thank you for this comment. We have mentioned the reference category as requested (see abstract, results and discussion section)

Comment: 3. The authors state that any disease-modifying treatment exposure was not associated with COVID-19 severity. However, a recently published study suggests that interferon antibodies are present in COVID-19 patients with a life-threatening course, especially in men and older patients (DOI: 10.1126/science.abd4585). Therefore, substituting interferon(-beta) may be beneficial regarding COVID-19 severitiy. If feasible, the authors could carefully comment on the 6 (4.8%) multiple sclerosis patients receiving interferon-beta having mild (and not life-threatening) COVID-19 (according to Suppl Table 1).

Response: Thank you again for this comment. While this is certainly an interesting aspect, we do not think that our data stemming from 6 patients treated with interferon beta preparations allows for an evidence-based or sufficiently informed comment on the potential role of autoantibodies against IFN-alpha2 and IFN-ω.

Comment: I thank the authors for their relevant scientific work.

Response: We thank the reviewer for the diligent work and constructive criticism.

Reviewer: 2

Comment: The authors report the findings on COVID-19 severity and mortality in MS patients investigated in a nation-wide Austrian registry. According to their results, outcome of COVID-19 is not modified by disease-modifying immunotherapy for MS. The methods are sound, the results are clearly presented, and the conclusions are justified. 

Response: Thank you for your positive assessment. 

Comment: 1) Methods. Please explain how diagnosis of COVID-19 was performed for the Austrian MS-COVID-19 registry.

Response: Thank you for this comment. Diagnosis of was defined either by a positive SARS-CoV-2 polymerase chain reaction [PCR] or a clinical diagnosis supported by i) a subsequent positive SARS-CoV-2 antibody test or b) a positive SARS-CoV-2 PCR in a close contact person). This was clarified in the methods section (p4). 

Comment: 2) Methods. Please insert a table outlining the MS-COV-risk score recently developed by the authors. This would help a lot to better understand the findings.

Response: Thank you for this suggestion. We have now included a table outlining the MS-COV-risk score as requested. 

Comment: 3) Table 1. please also show range for age

Response: We have added range for age as requested. 

Comment: 4) Page 6, punctuation after reference [12]

Response: Thank you for spotting this error. We have corrected the punctuation accordingly.

---

## [Decision Letter · Decision Letter 1]

14 Jul 2021

COVID-19 severity and mortality in multiple sclerosis are not associated with immunotherapy: insights from a nation-wide Austrian registry

PONE-D-21-12442R1

Dear Dr. Bsteh,

We’re pleased to inform you that your manuscript has been judged scientifically suitable for publication and will be formally accepted for publication once it meets all outstanding technical requirements.

Kind regards,

Orhan Aktas, M.D.

Academic Editor

PLOS ONE

Additional Editor Comments (optional):

Reviewers' comments:

Reviewer's Responses to Questions

**Comments to the Author**

1. If the authors have adequately addressed your comments raised in a previous round of review and you feel that this manuscript is now acceptable for publication, you may indicate that here to bypass the “Comments to the Author” section, enter your conflict of interest statement in the “Confidential to Editor” section, and submit your "Accept" recommendation.

Reviewer #2: All comments have been addressed

2. Is the manuscript technically sound, and do the data support the conclusions?

Reviewer #2: Yes

3. Has the statistical analysis been performed appropriately and rigorously? 

Reviewer #2: Yes

4. Have the authors made all data underlying the findings in their manuscript fully available?

Reviewer #2: Yes

5. Is the manuscript presented in an intelligible fashion and written in standard English?

Reviewer #2: Yes

6. Review Comments to the Author

Reviewer #2: All my comments have been addressed. The study adds important aspects to our understanding of the possible impact of immunotherapies on the outcome of MS patients in the pandemic era.

7. PLOS authors have the option to publish the peer review history of their article (what does this mean?). If published, this will include your full peer review and any attached files.

Reviewer #2: No

---

## [Editor Report · Acceptance letter]

19 Jul 2021

PONE-D-21-12442R1 

COVID-19 severity and mortality in multiple sclerosis are not associated with immunotherapy: insights from a nation-wide Austrian registry 

Dear Dr. Bsteh:

I'm pleased to inform you that your manuscript has been deemed suitable for publication in PLOS ONE. Congratulations! Your manuscript is now with our production department. 

Kind regards, 

on behalf of

Dr. Orhan Aktas 

Academic Editor

PLOS ONE